# Properties and Durability of Cement Mortar Using Calcium Stearate and Natural Pozzolan for Concrete Surface Treatment

**DOI:** 10.3390/ma15165762

**Published:** 2022-08-20

**Authors:** Jang-Hyun Park, Chang-Bok Yoon

**Affiliations:** 1Korea Institute of Future Convergence Technology, Hankyong National University, Anseong 17579, Korea; 2Department of Architectural Engineering, Seoil University, Seoul 02192, Korea

**Keywords:** calcium stearate, water repellent, natural pozzolan, diatomite

## Abstract

Applying a concrete surface treatment method (epoxy or primer) can prevent water from penetrating concrete through surface pores. However, if the concrete surface is damaged, the subsequent reconstruction can be expensive and time-consuming. Concrete that is resistant to internal and external water has been extensively developed and used to supplement the surface treatment method. Herein, we prepared specimens by mixing cement mortar with fatty-acid-salt-based calcium stearate attached to two natural pozzolanic materials—diatomite and yellow clay. The physical tests measured (1) the air content, (2) flow test, (3) compressive strength, and (4) activity Factor. Durability experiments were performed on (1) the contact angle, (2) chloride ion diffusion coefficient, and (3) water absorption test. The results revealed that the compressive strength of concrete decreased as the calcium stearate content increased. Furthermore, it was confirmed that the contact angle of the test piece using the pozzolanic substance and calcium stearate was twice as high. It was confirmed that the sand test specimen had the highest water absorption rate, and the DT3% had the lowest. (Sand%: 11.8 > OPC: 6.5 > DT3%: 2.4), the chloride diffusion coefficient also showed similar results. (Sand%: 12.5 > OPC: 8.4 > DT1%: 8.8)Due to its unique insolubility, calcium stearate retards hydrate formation when mixed alone and negates compressive strength loss when combined with pozzolanic mixtures rich in SiO_2_ and Al_2_O_3_. Furthermore, the ideal method for producing water-resistant cement mortar is to evenly disperse calcium stearate in the porous powder of cement mortar.

## 1. Introduction

The most representative factors for the degradation of concrete durability are carbonation, freezing and thawing, damage by chlorides, and the penetration of water through the concrete surface. The concrete surface appears to be dense macroscopically; however, penetration is possible because of the presence of numerous micropores on the material surface [1,2].

Considering that the micropores on the concrete surface allow the movement of water, various strategies have been developed to prevent the damage caused by water [3,4,5,6]. Methods for preventing water penetration into concrete surfaces can be classified according to their functions into surface coating, hydrophobic impregnation, pore blocking, and multifunctional surface treatments [7].

The surface coating forms a film on the concrete surface and inhibits the penetration of the deterioration factor [8,9,10].

Maravelaki-Kalaitzaki (2007) and Moradllo et al. (2008) [11,12] mentioned the technique of applying a penetrating water repellent to the surface without using a separate protective layer such as a urethane waterproofing or waterproofing sheet to increase the durability of concrete for the prevention of moisture-induced deterioration. Currently, the most effective technique to prevent water absorption is sealing the exposed concrete surface with a coating film, such as a rubber asphalt surface treatment. A surface treatment suppresses the internal rebar deterioration by protecting the concrete surface and controlling the water intrusion [13]. However, waterproof coatings that are limited to the surface may quickly deteriorate owing to climatic conditions or lack of user maintenance. When a small portion of coating deteriorates, the prevention of the overall water penetration of concrete is difficult [11,12,14,15].

When the surface protective layer is destroyed by such deterioration, moisture can easily permeate into the concrete, which can cause corrosion of the reinforcing bars and weaken the durability of the structure.

If resistance to moisture is provided inside the test piece to compensate for such a problem, it is possible to reduce the decrease in concrete durability due to the resistance to moisture inside, even if moisture permeates from the outside. In addition, it is possible to predict the extension of the life of the structure, and it is possible to take preventive measures against the time and cost required for repair [14].

In this study, in order to overcome the shortcomings of the existing concrete surface treatment technology and to develop water resistance inside the concrete, a metal salt-based calcium stearate powder having resistance to moisture was used. Calcium stearate reduces the permeation of chloride due to water absorption so that the durability of the specimen can be improved [16,17,18]. Calcium stearate has the property of being insoluble in water, and when added at the time of concrete compounding, bleeding phenomenon due to material separation can be predicted, so diatomite and yellow clay were used to solve the problem. Diatomite and yellow clay are porous as natural mineral powder materials, and after liquefying calcium stearate, they were prepared so that they could adhere to many voids and finished as an admixture material. Then a test specimen was prepared by changing the ratio according to the cement weight ratio.

We evaluate the physical properties (contact angle, water absorption, chloride ion diffusion coefficient, pore size, and distribution [via mercury intrusion porosimetry (MIP)]) and durability of each specimen at different curing days and analyze the results. Further, we examine the resistance of the mixture of porous natural pozzolanic materials and calcium stearate to water inside the cement mortar.

## 2. Materials

### 2.1. Diatomite

Diatomite is a light and porous pozzolanic material composed of the deposited shells of phytoplankton. The primary components of the shells are amorphous silica and opal, with a chemical formula of SiO_2_ nH_2_O. The content of the main component—SiO_2_—usually ranges from 70% to 80%. Diatomite also contains Al_2_O_3_, Fe_2_O_3_, CaO, and MgO. The particle size is generally 1 mm and has various shapes, depending on the phytoplankton species. Owing to its amorphous shape and porous structure, diatomite has unique physical properties such as high porosity, water permeability, surface area, chemical stability, water absorptivity, and heat resistance. Diatomite is mined from the Cenozoic strata in Pohang and is primarily used in South Korean agriculture [19]. Figure 1 shows the scanning electron microscopy (SEM) images of diatomite.

Because diatomite has several micropores, a low sintering temperature, and abundant domestic and foreign reserves, it can reduce cement manufacturing costs more effectively. Diatomite easily absorbs moisture with a spatial structure 5000 times finer than charcoal and can decompose bacteria by performing photocatalytic actions with its inherent TiO_2_ component. Diatomite can absorb liquids that are two to three times its own weight owing to the large pore volume it achieved upon silica absorption.

It is also reported to have high use value as an admixture material for concrete [20,21]. In this regard, there is a report [22,23] that the compressive strength and flexural strength of concrete were the best when diatomite and yellow clay were used instead of 10% of the cement mass. In recent years, calcined diatomite has been used as a concrete admixture to reinforce the durability of concrete and increase strength efficiency. Research on this calcined diatomite admixture is underway [24,25,26].

### 2.2. Yellow Clay

Yellow clay is an eco-friendly material with excellent far-infrared emissivity, deodorant performance, and humidity-control capacity. It primarily contains Al_2_O_3_ and Fe_2_O_3_. Its reserves are abundant enough to represent approximately 15% of the soil in South Korea, and it features the characteristics of natural pozzolans [27]. For use as a concrete admixture, yellow clay requires a firing process owing to the loss of compressive strength and drying shrinkage. The firing process is usually performed at 700–900 °C. 3CaO⋅2SiO_2_⋅3H_2_O (C-S-H) and 2CaO⋅Al_2_O_3_⋅SiO_2_⋅8H_2_O (C-A-S-H) crystals are formed, and as a result, bonds are formed between the materials, resulting in the expression of strength and high applicability as a cement admixture [24]. The activation of SiO_2_ and Al_2_O_3_ can overcome the above shortcomings by causing a pozzolanic reaction with Ca(OH)_2_ in the cement [28,29,30].

According to previous studies, the benefits of yellow clay continue to exist after high-temperature firing. The pozzolanic reaction reduces the porosity, which results in high chemical resistance to sulfuric acid and hydrochloric acid [31]. Figure 2 shows the SEM images of yellow clay.

### 2.3. Calcium Stearate

Calcium stearate is an efficient moisture-proof mixture derived from the reaction between stearic acid and limestone. A previous study reported that cement mortar and a paste mixed with calcium stearate significantly improved the capillary tube water absorption durability characteristics under non-static pressure [32]. Naeseroleslami, R., & Chari, M. N. (2019) analyzed the characteristics affecting SCC (Self-consolidating concrete) by incorporating CS to improve the short- and long-term impermeability of concrete, including SCC. As a result of the experiment, the incorporation of CS was much more effective than the SCM (Supplementary cementitious materials) material in the chloride transfer test, and the water-repellent layer formed along the capillary pores also significantly reduced the capillary water absorption. As a result of microanalysis, it was mentioned that CS mixing causes microcracks, and the compressive strength is reduced by reducing the density, but the strength loss can be compensated by lowering the W/B ratio or mixing silica fume [32]. However, it also recognized that physical properties, such as a reduction in compressive strength, were affected when using calcium stearate at 1% more than the concentration of mortar.

If the metal ions are divalent (M^2+^) or trivalent (M^3+^) instead of monovalent (M^+^), they are insoluble in water. When divalent and higher valent metal ions and fatty acids combine and react on the specimen surface, fatty acid metal salts are easily formed. Generally, metal ions become more insoluble as the oxidation number increases. Calcium stearate is insoluble because its solubility product in an aqueous solution is 3.61 × 10^−15^ [33]. Therefore, if calcium stearate is added during concrete mixing, the hydrophobicity of the cement mortar may occur. The impeded water transport in concrete due to the use of the hydrophobizing additive also inhibits the occurrence of efflorescence on concrete.

### 2.4. Other Materials

We used ordinary Portland cement (OPC, S Company, Seoul, Korea) in accordance with KS F 5201 [34]. We used the Korean washed sand, which meets the regulations of ISO 679, as the fine aggregate. Table 1 shows the chemical composition of OPC. Yellow clay produced from Gochang, Jeollanam-do, South Korea, and diatomite mined from and processed in Yeongilman, Gyeongsangbuk-do, South Korea, were used for the experiment. The chemical compositions of diatomite, yellow clay and calcium stearate are listed in Table 2 respectively. The specific gravity of yellow clay was 2.7, and its pH ranged from 5.0 to 7.0. The specific gravity of diatomite was 2.0–2.3, and its pH ranged from 9.0 to 10.0. Calcium stearate used in this study was a commercially available fine white powder (D Company, Seoul, Korea) and was neutral or weakly alkaline with a density lower than that of the cement. 

## 3. Methods

### 3.1. Experimental Plan

We prepared the cement mortar specimens for the experiment in accordance with KS L5105 [35]. We prepared the mortar in the following sequence:After slowly mixing the cement and sand, water was added and slowly mixed for 30 s at a speed of 140 ± 5 r/min.The mixer was stopped and restarted at a medium speed of 285 ± 10 r/min for 30 s.The mixer was stopped for 90 s before mixing was performed at 285 ± 10 r/min for 60 s.

### 3.2. Experimental Method

In the experiment, we measured the flow, compressive strength, and activity factor to evaluate the physical properties. In the durability evaluation experiment, we measured the water-repellent contact angle, water absorption test, and chloride ion diffusion coefficient. We conducted MIP to analyze the pore structure.

#### 3.2.1. Evaluation of Flow, Air Content, Compressive Strength, and Activity Factor

We measured the flow to examine the fluidity of the cement mortar in accordance with KS L 5105 [35], KS L 5111 [36], and ASTM A. C39 [37]. In addition, we measured the compressive strength at 7, 28, and 91 d using a universal testing machine. We measured the air content of the cement mortar in accordance with KS F 2421 [38].

For the experimental method, we liquefied calcium stearate powder using isopropylene as a solvent and sprayed it onto the sand, yellow clay, and diatomite to derive a mixing ratio with moisture resistance and to achieve uniform dispersion inside each specimen. We mixed calcium stearate with sand, yellow clay, and diatomite in a 1:1 weight ratio and naturally dried it for 1 d before use. Further, we added the dried powder at 1% or 3% of the cement weight to fabricate the cement mortar. Six types of specimens were prepared, including OPC (which was the control group). We fixed the water–binder (W/B) ratio at 50%. The cement to fine aggregate ratio was 1:3. We fabricated prismatic specimens with dimensions of 50 mm × 50 mm × 50 mm to test the physical properties and compressive strength. To measure the chloride ion diffusion coefficient and for use in the water penetration test, in order to measure the chloride ion diffusion coefficient, a cylindrical specimen with a size of ⌀100 mm × 200 mm was prepared, 70 mm × 70 mm × 20 mm specimens were prepared for the water absorption test. We demolded the cement mortar specimens after 24 h and immersed them in a water tank at 20 ± 2 °C for water curing for 91 d [35,36,37].

In the mortar flow test (KS L 5111), we filled a freshly mixed mortar inside a 70 mm × 100 mm × 50 mm mold installed on a round table with a 25.4 cm diameter. Upon turning the handle, the equipment rises to about 12.7 mm and falls to apply an impact. We repeated this 25 times. We then measured the diameter of the mortar spread caused by the impact.

We measured the air content according to the Korea Institute Standard on the method for testing the air content of fresh concrete by using the pressure method [38].

We performed the test using the water penetration test; wherein water is put into an air content meter equipped with a water inlet and drain port. We filled one-third of a container with the sample, evenly minced 25 times, and hit with a wooden mallet to remove the foam. Further, we added the sample repeatedly such that it slightly overflowed; we leveled the top surface using a ruler. We then attached the lid to the container and sealed it; we poured water through the inlet. We applied pressure using a pump. After 5 s, we sufficiently opened the valve to check the scale.

We evaluated the activity factor based on the compressive strength of the specimen at 91 d in accordance with KS L 5405 [39] using the following equation. The activity index was calculated by the calculation method of Equation (1):(1)$$As=\frac{C_{2}}{C_{1}}\times 100\left(\%\right),$$
where *As* is the activity factor (%), *C*_1_ is the compressive strength (MPa) of the OPC specimen, and *C*_2_ is the measured compressive strength (MPa) of the specimen mixed with the powder. The activity index is a value expressed as a percentage of the compressive strength ratio of a test mortar manufactured using an admixture and OPC to the compressive strength of a standard mortar manufactured with OPC.

#### 3.2.2. Contact Angle Measurement

The sessile drop method is a technique for determining the surface tension of a liquid. The equipment used for the experiment was a contact angle analyzer from Company E in South Korea that could measure the contact angle from 1° to 178° in a −50 to 150 °C temperature range and a 0% to 100% relative humidity range. In the sessile drop method, an arbitrary point that enables the most precise measurement and measures the coordinates and angle of the tangent is selected. It has a wide application range and can improve the reliability of calculation results because the process is simple, and there are no restrictions on the angle or density [40].

To measure the contact angle, we cut and dried the central part of the prismatic specimens; we then applied three 1 µL water droplets to the measurement surface, which was divided into nine sections. Figure 3 shows how the contact angle was measured using the sessile drop method [40,41,42,43]. Table 3 shows the water penetrability according to the contact angle.

In general, from the measured contact angle, the change in the monoatomic layer of the surface can be precisely analyzed, and the desired information can be obtained with reproducibility within a short period by using a simple analysis method. Table 3 shows the permeability according to the contact angle.

#### 3.2.3. Chloride Ion Diffusion Coefficient Measurement

Once curing was complete, we cut each ⌀ 100 mm × 200 mm cylindrical cement mortar specimen into four pieces with a thickness of 50 mm. We used the middle piece to evaluate the chloride ion diffusion coefficient inside the specimen [45]. For sufficient penetration of saturated calcium hydroxide (Ca(OH)_2_) into the specimen, we maintained a vacuum state using a vacuum pump and desiccator [46]. Table 4 shows the test voltages and durations for concrete specimens with normal binder content.

The external exposure test is the most accurate method for evaluating the chloride penetration resistance of concrete. However, because it requires considerable time, we evaluated the diffusion coefficient of concrete and its resistance to harmful ions using an acceleration method based on electrochemistry. In general, the ASTM C 1202 and NT Build 492 experimental methods are commonly used. In this study, we used the NT Build 492 method, a Nordic regulation commonly used to quantitatively evaluate the chloride ion diffusion coefficient of non-steady-state migration experiments, as an experimental method for chloride ion penetration resistance. Upon the completion of pretreatment, we constructed a chloride ion diffusion cell and measured the initial current value (l30V) by filling the anode with 0.3 NaOH aqueous solution and the cathode with 10% NaCl aqueous solution [46,47,48,49]. Further, we adjusted the actual applied voltage by determining the range of milliamperes according to the initial current value and conducted a test that used the potential difference by selecting the appropriate time according to the current. After completing the test, we divided the cross-section in the axial direction using a device and evenly sprayed 0.1 M silver nitrate (AgNO_3_) solution inside. We measured the discolored area to determine the chloride ion penetration depth. We cut the divided side of the specimen into seven sections at 10 mm intervals to measure the depth. We obtained the chloride ion penetration depth by averaging the measured depths and estimated the chloride ion diffusion coefficient (2).
(2)$$D_{\mathrm{nssm}}=\frac{RT}{zFE}\frac{x_{d}-\alpha \surd x_{d}}{t},\;for\;E=\frac{U-2}{L},\;\;\alpha =2\sqrt[{}]{\frac{RT}{zFE}}erf^{-}1\left(1-\frac{2Cd}{Co}\right),$$
where *D*_nssm_ is the non-steady-state migration coefficient (m^2^/s), *R* = 8.314 J/(K·mol) is the gas constant, *F* = 9.648 × 10^4^ J/(V·mol) is the Faraday constant, U is the absolute value of the applied voltage (V), *T* is the average of the initial and final temperatures in the anolyte solution (°C), *L* is the thickness of the specimen (mm), $x_{d}$ is the average value of the penetration depths (mm), *t* is the test duration (h), *Cd* is the chloride concentration when the color changed, and *Co* is the chloride concentration in the catholyte solution. Figure 4 shows the equipment used in the NT Build 492 measurement test [46,47,48,49].

#### 3.2.4. Water Absorption

We conducted the water penetration test at 28 d to measure the water absorption resistance of each specimen. We measured the weight of the penetrated water according to the water penetration resistance test method for cement-mixed polymer waterproof materials specified in KS F 4919 [50]. We treated the side of the specimen with epoxy paint for sealing such that water could contact and absorb the bottom surface of the specimen. We sufficiently dried the specimen inside the dryer until the start of the test.

Further, we removed the specimen and wiped the water on the surface. We measured the weight to determine the quantity of absorbed water. Absorption amount and absorption rate were calculated by the calculation method (3).
(3)$$Absorption\left(g\right)=W1-W0,$$

For the test method, the mass of the specimen before the absorption test was measured (*W*0) using a prismatic specimen of 70 mm × 70 mm × 20 mm at the age of 28 days. After that, the specimen was leveled and immersed to a depth of about 10 mm. After 24 h had elapsed, the specimen was taken out, the surface moisture was removed, and the surface was lightly wiped to measure the mass (*W*1) thereafter. Figure 5 shows a schematic diagram of the water absorption test.

#### 3.2.5. MIP

MIP is commonly used to examine the size and distribution of pores. We used it in this study to examine the pore structure of each specimen. For MIP, we used the cement mortar hardened for 91 d. For accurate analysis, we stopped the hydration for the prepared samples according to the curing time. We completely removed the residual moisture by drying it at 60 °C. We prepared specimens with dimensions of 5.0 mm × 2.5 mm and weights of 2–3 g. The equipment used for the experiment was an AutoPore IV mercury porosimeter from Micromeritics. We injected mercury at a maximum pressure of 430 MPa and an angle of up to 130°. After measuring the initial weight of the specimen, we derived a graph using the weight difference of mercury injected in low and high-pressure environments. Figure 6 shows the MIP system used [46].

## 4. Experimental Results

### 4.1. Evaluation of Physical Properties

Compared to the reference mortar, the specimens with calcium stearate (1% or 3% of the cement weight) and pozzolanic admixtures showed no significant difference in air content, although the air content slightly decreased as the contents of diatomite and yellow clay increased. When calcium stearate was directly sprayed onto the sand and then dried and stirred, the compressive strength was low. This might be due to the insufficient pore distribution of sand in relation to that of diatomite and yellow clay; thus, calcium stearate was separated during stirring. Lagazzo et al. reported that the air content of the cement mortar increases when sodium oleate is mixed with calcium stearate [51]. Because we used diatomite and yellow clay with a high fineness of 15,000 cm^2^/g in this study, the air content of the cement mortar appeared to have decreased. In addition, the results of this study were similar to those of a previous study [52] in that the addition of calcium stearate decreased the air content as it filled the internal pores of the mortar.

Owing to the pore structure and high fineness of diatomite, the specific surface area for moisture absorption increased, and the flow decreased as the diatomite and yellow clay contents increased. Table 5 lists the experimental results, and Figure 7 shows the measurement.

We measured the compressive strength at 7, 28, and 91 d and evaluated the activity factor of the specimen based on the results. We found that the compressive strengths of all specimens were lower than those of OPC. In relation to that of OPC, the expressed compressive strengths were 79% for YC1%, 73% for YC3%, 92% for DT1%, and 84% for DT3%. For the sand specimen with calcium stearate sprayed onto the fine aggregate, the compressive strength appears to have decreased because the separation of calcium stearate decreased the amount of generated initial hydrates because of the delay in the hydration reaction and bonding [53].

In the case of specimens in which calcium stearate was attached to diatomite and yellow clay, the compressive strength was also slightly lower than that of the reference mortar under the influence of calcium stearate, which was separated from the porous powder. However, as age increased, the strength gradually increased because SiO_2_ and Al_2_O_3_ included in the yellow clay and diatomite were activated, causing a pozzolanic reaction with Ca(OH)_2_ in the cement. The compressive strength differed slightly depending on the contents of SiO_2_ and Al_2_O_3_. Table 6 shows the mean and standard deviation after measuring the compressive strength three times for each curing time. We evaluated the activity factor based on the compressive strength of the specimen at 91 d in accordance with KS L 5405 [39]. Figure 8 shows the measured compressive strength and activity factors. The activity factor is a value expressed as a percentage of the compressive strength ratio of a test mortar manufactured using an admixture and OPC to the compressive strength of a standard mortar manufactured with OPC.

In the case of the sand specimen in which calcium stearate was attached to sand, stearate separated from the sand surface was not fixed inside the cement mortar. This resulted in a reduction in the compressive strength because of the delay in the hydration reaction owing to the unique insolubility of calcium stearate, similar to the result achieved when calcium stearate is mixed alone. However, the internal pores and reduced compressive strength were partly preserved through the combination with natural pozzolanic materials with high fineness and rich in SiO_2_ and Al_2_O_3_. It is judged that Ca(OH)_2_ of cement reacts with Al_2_O_3_ contained in diatomite and yellow clay to show the same level of initial strength as the standard mortar at YC1% and DT1%. At the compressive strength of 91 days, as the amount of calcium stearate increases, the compressive strength tends to decrease as compared with the reference mortar. However, it can be confirmed that the YC and DT test specimens show an increase in compressive strength as compared with the sand test specimen. Based on these compressive strength reduction results, we concluded that mixing calcium stearate affected the microstructure and showed negative results for internal particles, similar to what was confirmed in previous studies [53,54]. However, the diatomite and yellow clay specimens preserved the compressive strength relatively compared to the sand specimens, which is believed to be that SiO_2_ of the pozzolanic material produced calcium silicate (CSH) hydrate by the pozzolan reaction with the C_3_S and C_2_S hydration products of cement.

### 4.2. Contact Angle

A general technique used to check the surface characteristics of a solid specimen is to measure the angle of the droplet formed by dropping a droplet on a horizontal solid specimen by the sessile drop method [55,56,57]. In the experiment, we cut the central part of each prismatic specimen at 91 d and divided the cut surface into nine sections. We measured the contact angle, which directly indicates water resistance, three times and calculated the average. DT3% exhibited the largest contact angle, followed by YC3%, DT1%, YC1%, sand, and OPC. The contact angle was proportional to the calcium stearate content. Except for the sand specimen, the hydration products and pozzolanic powder with attached calcium stearate particles filled the pores, while the pozzolanic reaction was activated from the beginning of hydration owing to the presence of SiO_2_ and Al_2_O_3_ in yellow clay and diatomite. The OPC and sand specimens could not form contact angles. For the sand specimen, it appears that calcium stearate could not be attached to sand particles and thus separated via the formation of a non-uniform dispersion, and the bleeding phenomenon occurred owing to its light-specific gravity at the beginning of hardening. Figure 9 shows the preparation and measurement of the specimen. Figure 10 presents the contact angle measurement results.

### 4.3. Chloride Ion Diffusion Coefficient

The Faraday constant was 9.648 × 10^4^ J/(V·mol). U was the absolute value of the applied voltage (V), *T* was the average of the initial and final temperatures in the anolyte solution (°C), *L* was the thickness of the specimen (mm), $x_{d}$ was the average value of the penetration depths (mm), *t* was the test duration (h), *C_d_* was the chloride concentration when the color changed, and *C_o_* was the chloride concentration in the catholyte solution.

The chloride ion diffusion coefficient may vary depending on the pore structure because the pores may contain chloride ions and paths for their diffusion [58]. For the diffusion coefficient obtained from the chloride diffusion test at 91 d, OPC showed the highest value (8.4 × 10^−12^ m^2^/s), followed by sand (12.5 × 10^−12^ m^2^/s), YC1% (9.4 × 10^−12^ m^2^/s), and YC3% (10.3 × 10^−12^ m^2^/s). The diffusion coefficient was found to be 8.8 × 10^−12^ m^2^/s for DT1% and 9.6 × 10^−12^ m^2^/s for DT3%. Figure 11 shows the chloride ion diffusion coefficient results. The OPC confirmed the lowest chloride diffusion coefficient results. Based on the result of this reduction in compressive strength, it was judged that when calcium stearate was mixed, as confirmed in the previous study, it had an effect on the microstructure and showed a negative result on the internal particles [53].

As shown in the compressive strength, the relatively low results of diatomite and yellow clay compared to the sand specimen indicate that SiO_2_, which is abundant in pozzolanic materials, activated the reaction with the hydration product of cement.

### 4.4. Water Absorption Test

The water absorption test was based on ks F 4919 (Cement-Polymer Modified Waterproof Coatings) [50]. Figure 12 shows the state under test. Table 7 lists the weight change and water absorption. The water absorption of the YC1% and YC3% specimens were approximately half of those of the reference mortar. For the DT1% and DT3% specimens, the ratio decreased to one-third.

Among all the experimental specimen, the experimental sand specimen had the largest amount of absorption. As confirmed by the compressive strength reduction, the insoluble compounds generated through the reaction with calcium hydroxide in the cement could not densely fill the internal pores. CS forms a water-repellent layer along the capillary pores and does not form calcium silicate hydrates or pore-blocking precipitates [59]. The YC and DT specimens were confirmed to have less water absorption. It can be inferred that the separation of calcium stearate is less compared to the sand specimen, and the content fixed inside specimen is relatively high.

The same result as the previous research was obtained that the amount of water absorption can be greatly reduced when calcium stearate is mixed. Calcium stearate can reduce the permeability of concrete in non-hydraulic conditions. This observation indicates that providing resistance to water absorption along the pores reduces moisture migration through the capillary pores and retards the water saturation of the pores [53,60].

### 4.5. MIP

To examine the pore structure of each specimen, we performed an experiment using mercury intrusion at a maximum pressure of 430 MPa and a contact angle of 130°. We injected mercury into a small specimen and determined the pore size from the amount of mercury injected. We used acetone to stop the hydration of the specimens before analysis. We performed the experiment after completely drying the specimens at low temperatures to reduce thermal deformation.

The graph shows the size of the differential curve and the pore diameter range where the pore distribution can be known. It was found that the specimen mixed with calcium stearate had many large void distributions of 100 nm or more in preparation for OPC. Calcium stearate increased the total porosity of the cement paste, consistent with previous results of increasing the critical pore radius [61,62,63].

OPC and the specimens that used diatomite and yellow clay exhibited similar peaks in the 10–100 nm section, and the sand specimen showed a higher-intensity peak than the other specimens. The pores in the sections over 10–100 nm affected the strength; therefore, the internal structure was determined to be dense in the 10–100 nm section that corresponds to the capillary pore section. As the curing days increased, because diatomite and yellow clay contain abundant amounts of SiO_2_ and Al_2_O_3_, calcium silicate hydrate (C-S-H) peaks were formed in the capillary pore range as the pozzolanic reaction was activated. However, the finding that OPC exhibited the highest compressive strength indicates that the pores in the specimens that used diatomite and yellow clay were filled with hydration products, calcium stearate, and fillers that did not affect the hydration reaction. Figure 13 shows the MIP results.

## 5. Conclusions

In this study, we performed experiments to evaluate the water resistance of the cement mortar mixed with a hydrophobic metal-salt-based calcium stearate powder and natural pozzolanic materials—yellow clay and diatomite. The experimental study findings are as follows:

(1)It was confirmed that the compressive strength decreased as the mixing amount of calcium stearate increased. Compared with the reference mortar OPC, the lowest compressive strength was 21 MPa for the sand test specimen, and the highest compressive strength was 35 MPa for the DT 1% test specimen. This indicates that calcium stearate tends to delay the hydration of cement and reduce the amount of hydrate generated.(2)From the contact angle measurement result it was confirmed that the size of the contact angle increased as the mixing amount of calcium stearate increased. Among the experimental specimen excluding the reference mortar, the experimental sand specimen was measured at the lowest value at 34 and the highest measurement at DC 3% to 76. Considering that the angle of OPC is 20, it was found that the resistance to moisture is about 3.5 times or more.(3)Chloride ion diffusion coefficient diffusivity was measured to be highest at 12.5 × 10^−12^ m^2^/s for the sand specimens and the lowest at DT 1% to 8.8 × 10^−12^ m^2^/s. The specimens with pozzolanic admixtures and calcium stearate exhibited resistance to water. They also showed a decrease in the extent of penetration of migrating chlorides, resulting in a lower chloride ion diffusion coefficient than that of the reference mortar.(4)As a result of the water penetration test, the sand test piece lost resistance to water and showed the largest amount of water absorption (11.8 g). The DT 3% specimens were found to have the highest resistance to moisture with a moisture absorption of 2.4 g. In the diatomite and yellow clay specimens, calcium stearate adhering to pores on the surface of the particles did not separate and was located inside the specimen and was judged to have high moisture resistance. From this, it can be inferred that calcium stearate is resistant to water in the non-hydraulic state (natural state).(5)The MIP test results, which are similar results to the compressive strength, were shown. It was shown that OPC, YC3%, and DT3% had a large void distribution in the 10–100 nm fine section, and conversely, the sand specimen had a large void distribution of 10^3^–10^6^. A large amount of SiO_2_ in diatomite and yellow clay and ocher produces calcium silicate (CSH) hydrate is due to the pozzolan reaction with C_3_S and C_2_S hydration products of cement.

## Figures and Tables

**Figure 1 materials-15-05762-f001:**
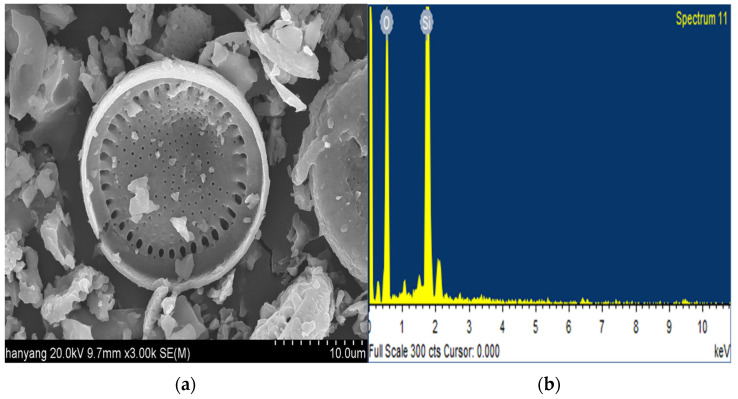
Scanning electron microscopy (SEM) images of diatomite at (**a**) ×3000 magnification; (**b**) EDX spectrum.

**Figure 2 materials-15-05762-f002:**
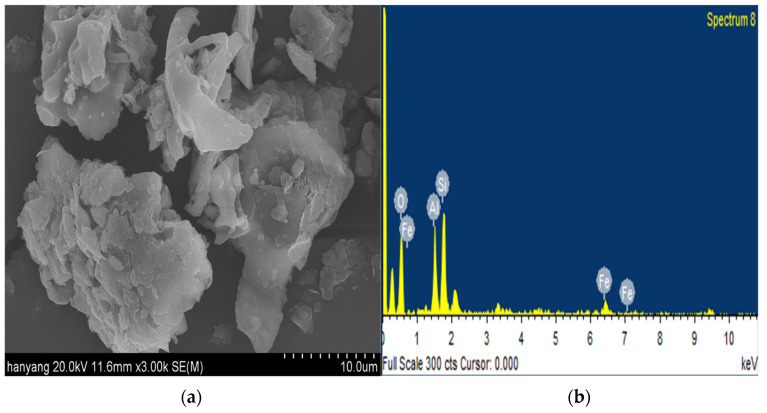
Scanning electron microscopy (SEM) images of yellow clay at (**a**) ×3000 magnification; (**b**) EDX spectrum.

**Figure 3 materials-15-05762-f003:**
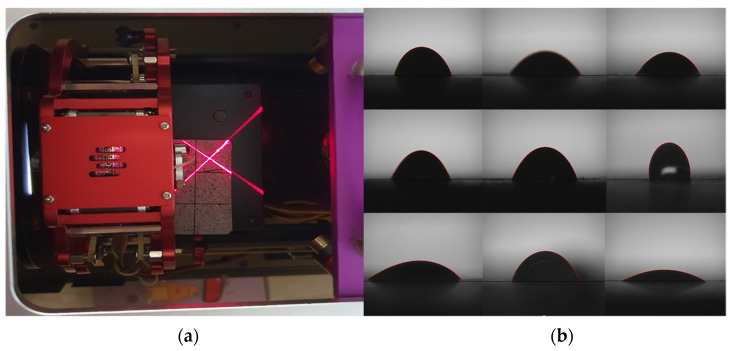
Measurement of the contact angle using the sessile drop method. (**a**) Sessile drop measuring device; (**b**) contact angle measurement result.

**Figure 4 materials-15-05762-f004:**
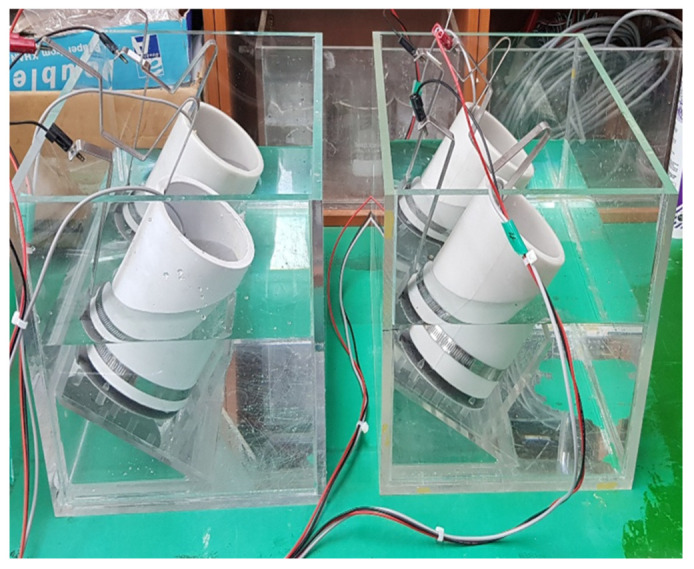
NT Build 492 measurement test equipment.

**Figure 5 materials-15-05762-f005:**
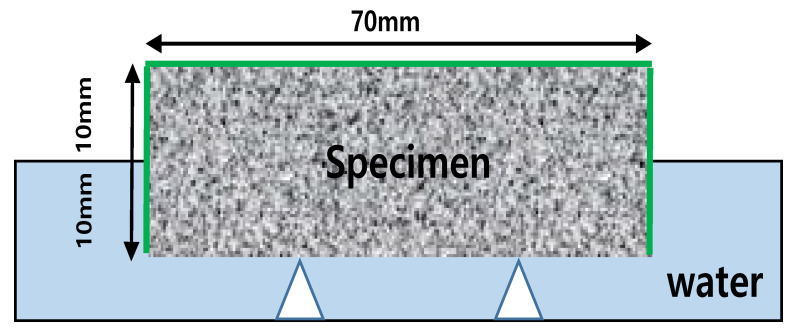
Water absorption test equipment.

**Figure 6 materials-15-05762-f006:**
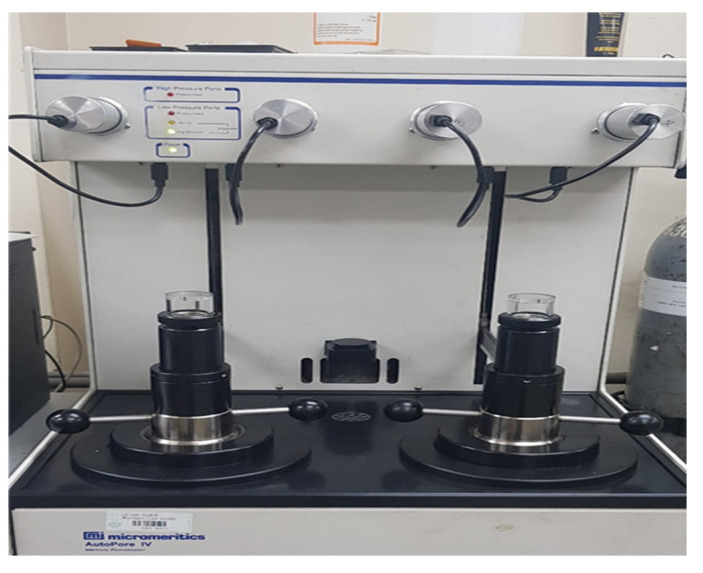
Mercury intrusion porosimetry (MIP) test setup.

**Figure 7 materials-15-05762-f007:**
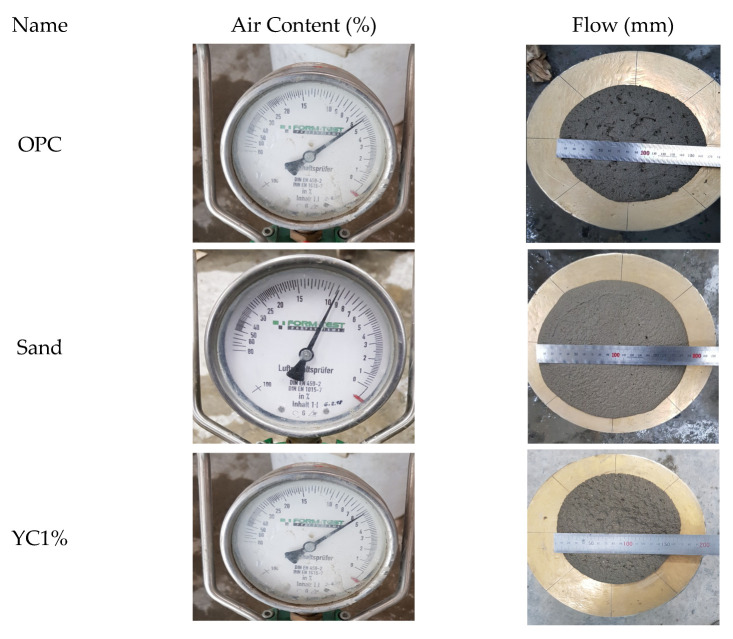
Measurement of the physical properties.

**Figure 8 materials-15-05762-f008:**
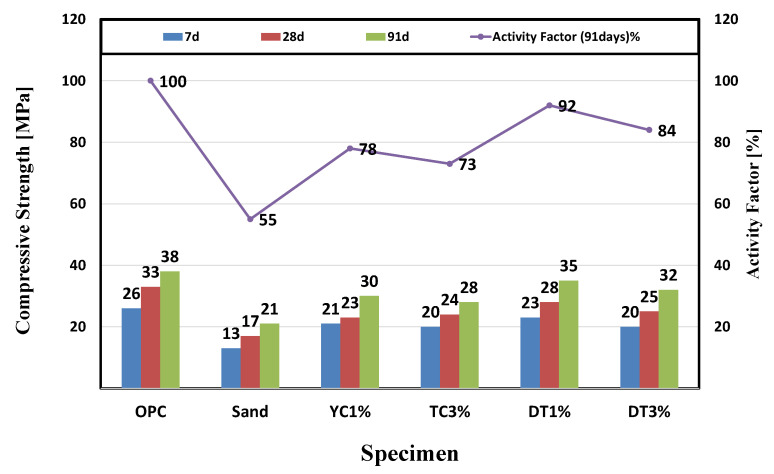
Measurement of Compressive strength and activity factor of the specimens.

**Figure 9 materials-15-05762-f009:**
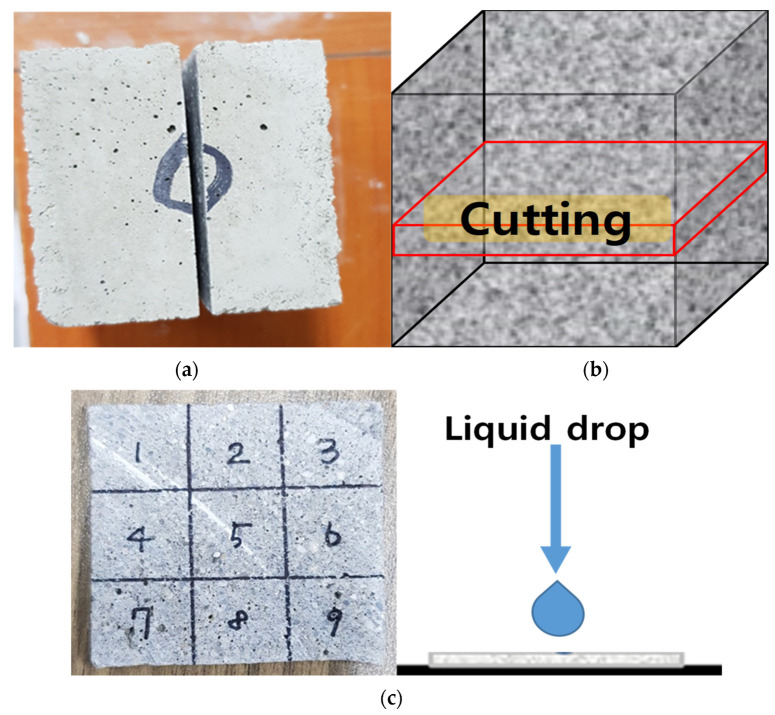
Specimen preparation and measurement. (**a**) Divided specimen; (**b**) end test specimen; (**c**) contact angle [46].

**Figure 10 materials-15-05762-f010:**
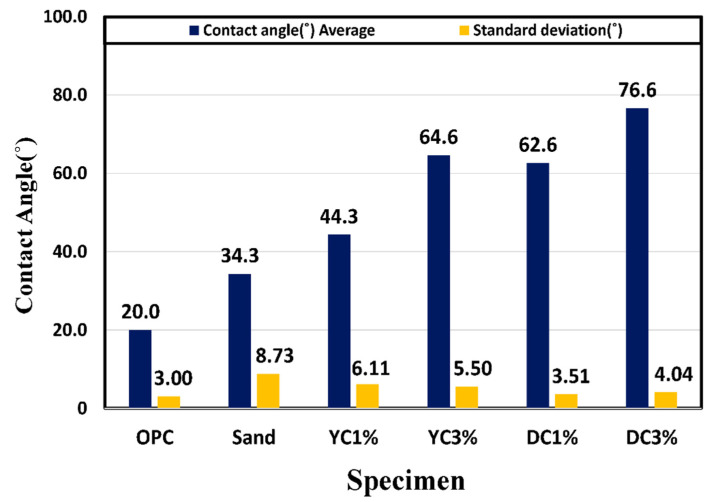
Measurement of contact angle (°) (91 d).

**Figure 11 materials-15-05762-f011:**
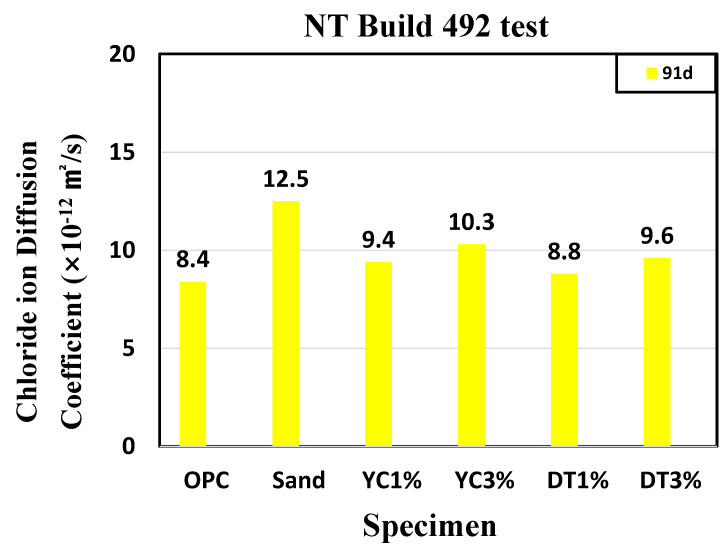
Results of the NT Build 492 test (91 d).

**Figure 12 materials-15-05762-f012:**
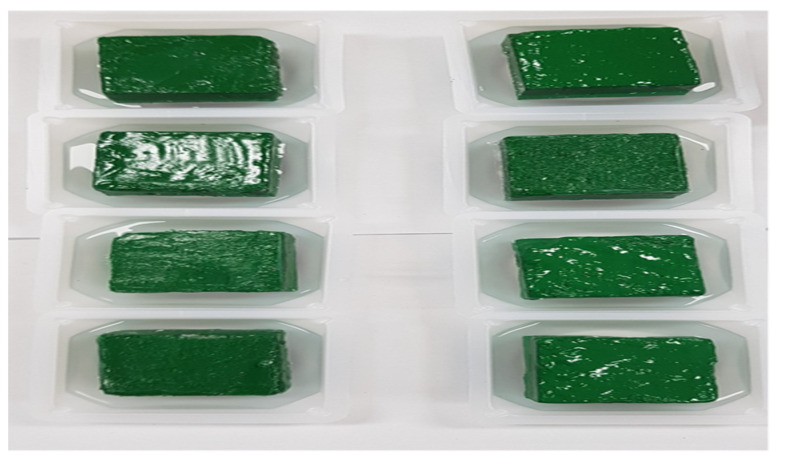
Measurement of Water absorption test (91 d).

**Figure 13 materials-15-05762-f013:**
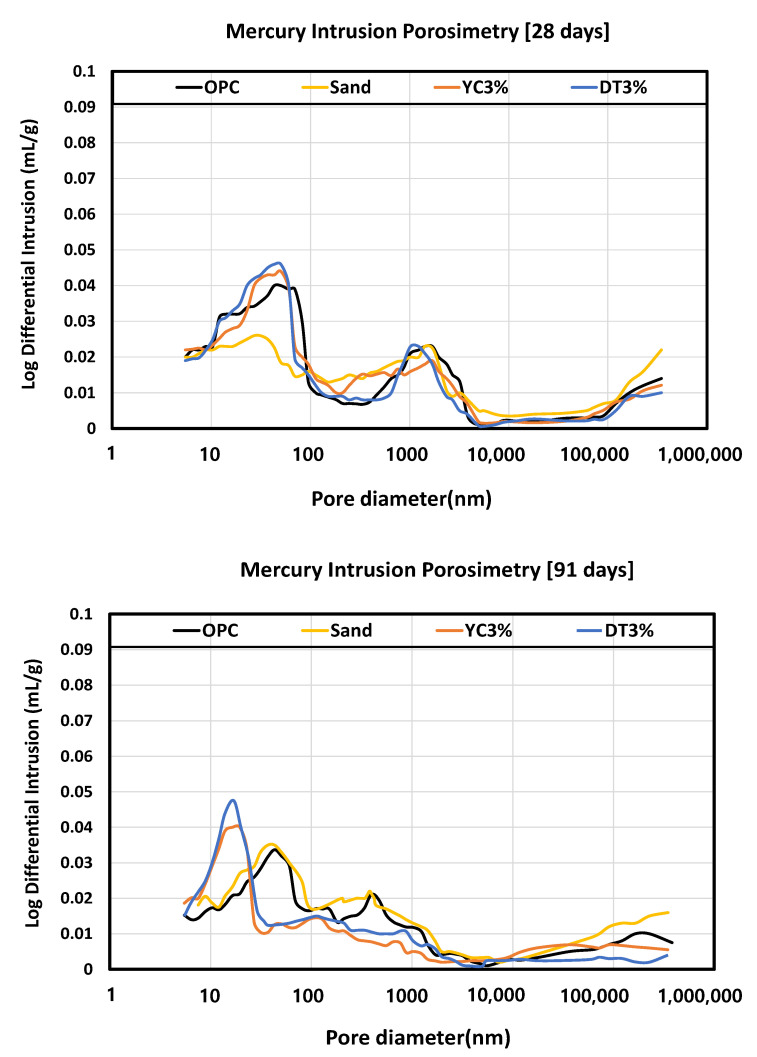
Measurement of MIP.

**Table 1 materials-15-05762-t001:** Chemical composition of binder materials (OPC, Yellow clay, diatomite, calcium stearate).

Name	Chemical Composition (%)								
	SiO_2_	Al_2_O_3_	Fe_2_O_3_	CaO	MgO	SO_3_	K_2_O	Etc. /Lg. loss	LOI
OPC	19.29	5.16	2.87	61.68	4.17	2.53	0.89	3.41	2.3
**Name**		**Chemical Composition (%)**							
		**SiO_2_**	**Al_2_O_3_**	**Fe_2_O_3_**	**MgO**	**CaO**	**K_2_O**	**Etc.** **/Lg. Loss**	
Yellow clay		42.9	37.0	6.4	1.1	1.3	0.8	10.5	
Diatomite		78.5	10.1	2.8	2.1	1.9	1.8	2.8	
**Name**	**Chemical Composition (%)**								
	**Chemical** **Formula**		**Density** **(g/cm^3^)**		**pH**	**Melting** **Point**	**Molecular** **Weight** **(g/mol)**		
Calcium stearate	C_36_H_70_CaO_4_		1.08		7–9	147–149	607		

**Table 2 materials-15-05762-t002:** Mix proportion of the cement mortar.

Name	W/B (%) *	Unit Weight (kg/m^3^)					
		Cement	Water	Sand	CS *	YC *	DT *
OPC *	50%	510	255	1530	-	-	-
Sand	50%	510	257.5	1530	5.1	-	-
YC1% *	50%	510	260.1	1530	5.1	5.1	-
YC3% *	50%	510	270.3	1530	15.3	15.3	-
DT1% *	50%	510	260.1	1530	5.1	-	5.1
DT3% *	50%	510	270.3	1530	15.3	-	15.3

* W/B: water–binder ratio, CS: calcium stearate, YC: yellow clay, DT: diatomite, OPC: ordinary Portland cement, 1% and 3%: proportion of dried powder to the weight of the cement.

**Table 3 materials-15-05762-t003:** Water penetrability according to the contact angle [40,41,42,43,44].

Surface Contact Angle	Penetrability
>130°	High repellency
110–130°	repellency
90–110°	Slight wetting
30–90°	Pronounced wetting
<30°	Complete surface wetting

**Table 4 materials-15-05762-t004:** Test voltage and duration for the concrete specimen with the normal binder content [46,47,48,49].

Initial Current I_30 V_ (with 30 V) (mA)	Applied Voltage U (after Adjustment) (V)	Possible New Initial Current I_0_ (mA)	Test Duration (h)
I_0_ < 5	60	I_0_ < 10	96
5 ≤ I_0_ < 10	60	10 ≤ I_0_ < 20	48
10 ≤ I_0_ < 15	60	20 ≤ I_0_ < 30	24
15 ≤ I_0_ < 20	50	25 ≤ I_0_ < 35	24
20 ≤ I_0_ < 30	40	25 ≤ I_0_ < 40	24
30 ≤ I_0_ < 40	35	35 ≤ I_0_ < 50	24
40 ≤ I_0_ < 60	30	40 ≤ I_0_ < 60	24
60 ≤ I_0_ < 90	25	50 ≤ I_0_ < 75	24
90 ≤ I_0_ < 120	20	60 ≤ I_0_ < 80	24
120 ≤ I_0_ < 180	15	60 ≤ I_0_ < 90	24
180 ≤ I_0_ < 360	10	60 ≤ I_0_ < 120	24
I_0_ ≥ 360	10	I_0_ ≥ 120	6

**Table 5 materials-15-05762-t005:** Air content and flow of the specimens.

Name	Air Content (%)	Flow (mm)
OPC *	5.6	165
Sand	9.2	190
YC1% *	5.9	170
YC3% *	5.5	160
DT1% *	6.0	175
DT3% *	5.2	165

* OPC: ordinary Portland cement, YC: yellow clay, DT: diatomite, 1% and 3%: proportion of dried powder to the weight of cement.

**Table 6 materials-15-05762-t006:** Measurement of the physical test.

Name	Compressive Strength (MPa)														
	7 d					28 d				91 d					
	1st	2nd	3rd	Average	* SD	1st	2nd	3rd	Average	* SD	1st	2nd	3rd	Average	* SD
OPC	25	25	28	26	1.7	35	31	33	33	2.0	38	39	36	38	1.5
Sand	16	10	13	13	3.0	15	20	16	17	2.6	16	25	22	21	4.5
YC1%	23	18	22	21	2.6	25	22	22	23	1.7	32	29	29	30	1.7
YC3%	18	20	23	20	2.5	21	24	25	24	2.0	32	29	29	28	1.7
DT1%	24	24	22	23	1.1	28	29	27	28	1.0	34	34	36	35	1.1
DT3%	22	19	20	20	1.5	23	25	27	25	2.0	32	34	30	32	2.0

* SD: Standard deviation.

**Table 7 materials-15-05762-t007:** Measurement of the Water absorption test (91 d).

Name	Dry (g)	Wet (g)	Absorption (g)	Normalized ValueOPC (%)
OPC	118.9	125.4	6.5	100
Sand	117.3	129.1	11.8	181
YC1%	119.5	122.9	3.4	52.3
YC3%	118.4	121.4	3.0	46.1
DT1%	119.1	122.4	3.3	50.8
DT3%	120.8	123.2	2.4	37.0

## Data Availability

Data is contained within the article.

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
