# Peer review of "Properties and Durability of Cement Mortar Using Calcium Stearate and Natural Pozzolan for Concrete Surface Treatment"

_materials, 2022, doi:10.3390/ma15165762_

Round 1

Reviewer 1 Report

The authors aimed to investigate the influence of calcium stearate associated with diatomite and yellow clay on the permeability to water and chloride ions of mortars. In my opinion, the manuscript, including language, methodology, results, figures/tables, requires major revision.

As there is no number of lines, I will indicate the changes by chapter and paragraph.

Abstract

The abstract must indicate the content/proportion of materials.

Authors must indicate absolute values. It may be indicated between parenthesis.

“ When other characteristics were measured, the specimens that used pozzolanic materials and calcium stearate exhibited at least twice as large contact angles (), approximately a third of the water penetration (), approximately half the chloride ion diffusion coefficients () than the reference mortar …”

What you mean with “…negates compressive strength reduction…” and “…porous powder…”?

Introduction

p. 2 – “Because micropores on the surface caused by the material properties of concrete allow the movement of water, studies have been conducted to prevent the damage caused by water.”

That sentence doesn't make sense.

There is no fluidity in the intro. The authors suddenly cite diatomite and yellow clay as potential pozzolans, as they can store calcium stearate because they are porous. The introduction should contextualize the reason for this work, that is, where did you get this idea from? Why not use common reactive pozzolans to reduce micropores by pozzolanic activity (forming additional CSH)?

2. Materials

p. 2 – “Because diatomite has many micropores, a low sintering temperature, and abundant domestic and foreign reserves, diatomite can reduce cement manufacturing costs when compared with other porous ceramic materials, such as Al2O3, ZrO2, SiC, and Si3N4.”

Why comparing diatomite with alumina or zirconia-based ceramics? They are not pozzolanic materials.

Figure 1 and 2 – SEM micrographs must include scale bar, and magnification in the caption. Also, Figures 1-4, 6-8, 10, 12 e 13 must be divided into a), b), c) and so on.

2.3 Calcium stearate

p. 2 – “Thus, the fatty acid metal salt of Fe3+ is expected to be better deposited on the cement mortar surface than the less insoluble Ca2+ [15]. Therefore, if calcium stearate is added during the mixing of concrete, the hydrophobicity of the cement mortar is expected to occur because of its unique insolubility.”

Where does Fe3+ come from? This paragraph is confusing.

The authors quote concrete while producing mortar. Wouldn't becoming more hydrophobic disrupt the hydration process?

Tables 1 to 4 should be grouped into a single one. The equipment and procedure used to determine the chemical and physical properties of raw materials were not cited.

Method

Why introduce water first? Normally the solids are dry mixed for homogenization, with the addition of water gradually. Thus, hydration begins homogeneously, as does the solubilization of cement particles.

“ … to derive a mixing ratio with resistance to moisture …” What does this means?

In Table 5, the mix proportion must be expressed as a percentage (preferably by mass).

Did the authors calculate the correct w/b (%)? Sounds like a lot of water at 50% w/b. See if my calculations are correct. Your data would imply a water/solids ratio of 1, while it normally comprises 0.5-0.6 in practice.

hypothetical values

water

1

solids

1

binder (water+solids)

2

water/binder %

50%

water/solids

1

3.2 Experimental methods

Specify MIP. What is activity factor and how is it measured?

The samples sizes for compressive strength were not cited.

The authors evaluate the flow by flow table, slump test, what equipment was used? Minimum information is recommended rather than just citing the Standard.

Table 6 is not mentioned in the text.

3.2.5 …

“The specimens were prepared with dimensions 5 × 5 × 5 mm, and their weights were between 2 and 3 g.”

If I'm not mistaken, with a volume of 25 mm³ (0.125 cm³) and weight between 2 – 3 g, the apparent density will be in the range of 16 to 24 g/cm³ (higher than lead).

4.1 Evaluation …

“To evaluate the physical properties, the air content, flow, compressive strength, and activity factor were measured and evaluated in accordance with KS L 5105 [17], ASTM, A. C39 [18], KS F 2421 [19], and KS L 5405 [20], respectively”

This sentence must be revised. In fact, the entire manuscript should be thoroughly proofread for language. Also, the standards have already been previously cited.

“Because diatomite and yellow clay with a high fineness of 15,000 cm2/g were used, the air content of cement mortar appears to have decreased in this study. In addition, the obtained results were similar to the results of a previous study [33] in that the addition of calcium stearate decreases the air content by filling the internal pores of the mortar.”

If calcium stearate is filling the internal pores, it means they weren't bonded to the diatomite and clay. The authors could have taken micrographs of diatomite after spraying the solution with calcium stearate. The authors should also have evaluated compositions with diatomite and yellow clay without calcium stearate for comparison.

Fig. 8 – The images presented do not bring any contribution. Table 8 should be replaced by a figure.

“The compressive strength was measured at 7, 28, and 91 d, and the activity factor of the specimen was evaluated based on the results. The compressive strength at 7 d was 24, 13, 22, 29, 23, and 20 MPa for the OPC, sand, YC1%, YC3%, DT1%, and DT3% specimens, respectively. The compressive strength at 28 d was 32, 17, 29, 26, 28, and 25 MPa for the OPC, sand, YC1%, YC3%, 28 DT1%, and DT3% specimens, respectively. The compressive strength at 91 days was found to be 38, 21, 30 , 28, 35, and 32 for the OPC, sand, YC1%, YC3%, DT1%, and DT3% specimens, respectively.”

It is not necessary to inform the absolute values side by side, as Fig. 9 already illustrates this. By the way, for what age was the activity factor measured?

Compressive strength is substantially reduced with the addition of diatomite and yellow clay associated with calcium stearate. We can conclude if in fact calcium stearate is the main responsible for this since there are no samples with only diatomite and clay (without CS). Furthermore, the authors did not investigate whether calcium stearate remains bound to pozzolans. This is very important as CS can disturb the hydration of cement particles due to its hydrophobic character.

Figure 9 and Table 9 show the same data. There is no need for both. Figures are often preferred.

Table 10 must be converted into a graph, with only the mean values ​​and standard deviation.

4.3 … the explanation about the test procedure, equations, etc. must be in the section Methods.

Unfortunately, analyzing the author's data, we cannot conclude whether calcium stearate is really useful and contributes to reducing the penetration of chloride ions and water, as it was only evaluated together with diatomite and yellow clay. Perhaps diatomite and yellow clay alone can improve/decrease the permeability of mortars without the help of CS, since both are reactive pozzolans.

Author Response

We would like to thank the reviewers for the thorough review of this manuscript and for the thoughtful comments and constructive suggestions, which help to improve the quality of the manuscript.

Reviewer 2 Report

In this study, specimens were prepared by mixing cement mortar with a fatty acidsalt-based calcium stearate attached to natural pozzolanic materials diatomite and yellow clay.The results showed that the compressive strength of the concrete decreased as the calcium stearate content increased. When other characteristics were measured, the specimens that used pozzolanic materials and calcium stearate exhibited at least twice as large contact angles, approximately a third of the water penetration, approximately half the chloride ion diffusion coefficients than the reference mortar, and multiple 10-100 nm micropores.

Because of its unique insolubility, calcium stearate delays the formation of hydrates when mixed alone and negates compressive strength reduction through a combination with pozzolanic admixtures rich in SiO2 and Al2O3.

In this study, experiments were performed to evaluate the water resistance of cement mortar mixed with hydrophobic metal salt-based calcium stearate powder, and natural pozzolanic materials yellow clay and diatomite.

Bearing in mind the entirety of the article, I support and request the possibility of publishing the article in the current form.

Author Response

Thank you so much for considering my paper.

We would like to thank the reviewers for the thorough review of this manuscript and for the thoughtful comments and constructive suggestions, which help to improve the quality of the manuscript.

Keep healthy.
Thank you and best regards.
Yours sincerely,
Chang Bok Yoon

Reviewer 3 Report

The paper materials-180195 investigates diverse fine concrete formulations including calcium stearate as hydrophobic agent for achieving higher durability performance, in combination with pozzolanic reactive natural materials to tune the pore size distribution and the mechanical response.

The topic is compliant with the journal and the idea behind the research possess a sufficient degree of novelty.

The paper presents major flaws that should be thoroughly addressed by the Authors. First, the quality of English language is not adequate and the text requires a revision, possibly by a native speaker.

The introduction is absent. The research problem is not framed and no adequate references to the existing literature are mentioned. The introductory section must be enlarged significantly to encompass all the background, as the issue of chloride penetration in concrete and related provisions have been under investigation for a very long time.

With regard to the results, how many specimens were tested in compression? Apart from the flow measurements, how was the workability quality of the mortars? Indeed, the Authors mention that diatomite and clay possess extremely high fineness, whereas the water content was kept constant and no water reducer was adopted.

Page 10: compressive strength values should be reported in a Table and not listed in the text, for the sake of clarity and readability.

Statements in page 10 (e.g. "the compressive strength appears to have decreased because the separation of the calcium stearate decreased the generation of initial hydrates by delaying the hydration reaction and delayed bonding) are neither supported by instrumental evidence nor by other studies in the literature. When discussing about microstructural effects, in-depth investigations are mandatory (like, e.g. SEM).

Again "However, as age increased, the strength gradually increased because the SiO2 and Al2O3 included in the yellow clay and diatomite were activated, causing a pozzolanic reaction with Ca(OH)2 in the cement". As cement is in all the mixes, the strength gain is expected and obvious. Indeed, it is evident even in the reference mixes, without pozzolanic materials.

In figure 9 (and elsewhere through the manuscript) no reference to the standard deviation values is reported. Data fluctuation is indeed as important as the mean values. The Authors are requested to clarify, for each experiment, how many repetitions were conducted and to explicitly report on the data scattering.

At the beginning of Sections 4.2, 4.3 and 4.4 the description of the test procedure and the pictures of the setup are redundant and misplaced.

In the conclusions, "When calcium stearate was added, the compressive strength slightly decreased". The word "slightly" is not objective. Indeed, they observed also significant reductions in strength.

Concluding remark: Despite several investigation techniques are used, the scientific rigour of the whole manuscript is limited, as the discussion (which is often dealing with microstructural considerations) is not supported by any specific microstructural investigation. No information about data scattering is provided, impairing the robust readability of the results. At it is, to my opinion I regret to state that the paper does not comply with the scientific standards of the journal "Materials".

Author Response

We would like to thank the reviewers for the thorough review of this manuscript and for the thoughtful comments and constructive suggestions, which help to improve the quality of the manuscript.

Thank you so much for considering my paper.
It was revised to reflect the review items of the reviewer to the maximum extent.

Reviewer 4 Report

The development of hydrotechnical, as well as underground construction is associated with the necessity of ensuring the tightness of building objects and increasing their resistance against chemical corrosion. Among several factors that affect the water tightness and corrosion resistance of concrete, the following must be listed: a) the type of cement use and its phase composition, b) the rate of chemical reaction in a cement hydration process and the rate of hardening, c) the sizes and continuity of capillaries and microcapillaries in a hardening slurry, d) the type of number of mineral additives used in the cement, e) capillaries and capillary vacuums at the aggregate-cement slurry interface, f) the use of special chemical dopants in the cement or coating concrete with chemicals that inhibit water penetration through the concrete and g) the conditions of care for the concrete. Some of these factors were the subject of the reviewed work. This paper mainly concentrated on determining the influence of calcium stearate addition and natural pozzolanic materials diatomite and yellow clay on the water tightness and resistance of cement mortar in a solution of chlorides. Both the writing and the overall quality of this experimental work are acceptable. The title is an adequate reflection of the substance of the article, and the results will certainly be valuable to researchers working in the field. The manuscript can be recommended for publication after the authors address the following major concerns:

1.Introduction: The introduction of this work is presented sparingly and consequently does not contain enough information on the current state of research concerning water tightness and resistance of concretes against chemical corrosion with specific inclusion of the cement slurry destruction process mechanism in an environment consisting of chlorides. Furthermore, it is necessary to rewrite the goal of this work, because it is not very precise.

2. Subsection 2.1: There is no reference to Fig. 1 in the main body of the manuscript. Moreover, a magnification scale should be included in this figure.

3. Subsection 2.2: There is no reference to Fig. 2 in the main body of the manuscript. Moreover, a magnification scale should be included in this figure.

4. Subsection 2.3: At the end of this subsection another sentence should be added: The impeded water transport in the concrete with the hydrophobizing additive also inhibits the occurrence of efflorescence on the concrete.

5. Subsection 4.1: The sentences "The compressive strength ...., respectively." Should be removed, because this data is presented in Table 9 and Figure 9.

6. Subsection 4.3: When interpreting the results of chloride ion diffusion coefficient measurements, did the authors take into account the fact that in the studied samples reactions between the chloride ions and cement slurry phases will take place, which will decrease ion penetration and thus the measured diffusion coefficient will be lower?

7. In this work, there is a lack of a deeper discussion of results in the context of e.g. the influence of pore size in cement mortars on their mechanical strength.

Author Response

(The authors gave the same response as above.)

Reviewer 5 Report

            This research has discussed about the usage of calcium stearate and natural pozzolan in mortar. Lot of improvement is required in the area of abstract and introduction part. Hence, article can be accepted after a major revision.

1. In abstract, there is no experimental procedures are clearly mentioned. What kind of tests are performed?

2. There is no clear mention about how many replacement are done? Either concrete or mortar?

3. Authors are requested to rewrite the abstract entriely. In the order of objective of work, experimental work, quantitative results

4. Article heading deals with cement mortar but the introduction is about concrete. Kindly make introduction about cement mortar subjected to durability properties

5. Only 4 literatures in introduction? Add more literatures related to studies.

 6. Research gap and objective of present study is not clear. It requires drastic improvement

7. Given a detailed introduction about the material going to use in this study

8.make table 1 to 4 in single table

9. how water to binder ratio is fixed?

10. In results and discussion,.most of experimental part are discussed as experimental results there is no proper scientific discussion with reference are there.

11. Section 4.2 & 4.4 there is no reference?

12. Section 4.3 remove the code name in brackets

13. Conclusion should reflect the experimental work done with specific output. Conclusion needs improvement

14. General comments: several sentences are too long and it is requested to use simple English.

Good luck

Author Response

(The authors gave the same response as above.)

Round 2

Reviewer 1 Report

There are methodological errors, and the revised article has changed little from the first version. The manuscript also lacks organization. I do not recommend publishing.

Author Response

(The authors gave the same response as above.)

Reviewer 3 Report

My comments were only partially considered and only minor changes were brought to the manuscript, which still addresses the problem in a superficial manner.

The introduction remained unchanged, despite my major remark about the lack of adequate references to frame the problem and explain the innovative value added of this contribution. This remark was ignored.

The readability of the paper is improved, however, many considerations and discussions are still not supported by adequate analytical investigation, like the microstructure of the conglomerates. I understand that the research is likely in progress and further studies will be performed, however, some in-depth analyses should be hereby reported.

I regret to say that, in my opinion, the improvements are not sufficient to reach the soundness and the rigour requested for a scientific publication in this Journal.

Author Response

(The authors gave the same response as above.)

Reviewer 4 Report

The authors addressed the comments and improved the quality of the manuscript. The revised version can be recommended for publication.

Author Response

(The authors gave the same response as above.)

Reviewer 5 Report

Add reply to individual comments from each reviewer.

Author Response

We would like to thank the reviewers for the thorough review of this manuscript and for the thoughtful comments and constructive suggestions, which help to improve the quality of the manuscript.

We'll send you the consolidated comments from each reviewer as requested.
We have reflected all the corrections requested by each reviewer, and responses to the corrections are also included.

Thank you so much for considering my paper.

It was revised to reflect the review items of the reviewer to the maximum extent.

Keep healthy.

Thank you and best regards.

Yours sincerely,

Chang Bok Yoon
